# Second Law and Its Amendment: The Axiom of No-Reversible Directions Revisited

**DOI:** 10.3390/e25081226

**Published:** 2023-08-17

**Authors:** Wolfgang Muschik

**Affiliations:** Institut für Theoretische Physik, Technische Universität Berlin, Hardenbergstr, 36, D-10623 Berlin, Germany; muschik@physik.tu-berlin.de

**Keywords:** balance and constitutive equations, dissipation inequality, second law, process directions, no-reversible-direction axiom, Liu relations

## Abstract

A toy model is used to describe the following steps to achieve the no-reversible-direction axiom in a tutorial manner: (i) choose a state space results in the balance equations on state space which are linear in the process directions, (ii) avoid a reversible process direction that cannot be generated via a combination of non-reversible ones, (iii) process directions that are in the kernel of the balance equations and do not enter the entropy production. The Coleman–Mizel formulation of the second law and the Liu relations follow immediately.

## 1. Preface

This paper is a short-note tutorial on a well-discussed topic of the material theory, specifically examining the following question: How do we achieve compatibility for constitutive equations with the second law formulated via a dissipation inequality? This classic problem is resolved here using a toy model, which allows for elucidating the steps, whereas the original literature is often complicated because of the chosen material and it lacking clear definitions of the used concepts. Consequently, nothing is really novel in this revisit, except for the tutorial-based view of the definitions and concepts, making the amendment of the second law more familiar.

## 2. Introduction

An often heard statement is “The engineer (and also the physicist) knows the material under consideration, why do we need a constitutive theory?”. Often, this statement is true, and this engineer can be happy because they know the corresponding constitutive equation describing his material under inspection, e.g., a non-perfect gas.
(1)$$\begin{array}{ccc} pV & = & nRT+\frac{n^{2}}{V}\left(B_{0}RT-A_{0}-\frac{C_{0}}{T^{2}}\right)+\\ & & +\frac{n^{3}}{V^{2}}\left(bRT-a\right)+\frac{n^{6}}{V^{5}}a\alpha +\frac{n^{3}}{V^{2}T^{2}}\left(1+\frac{n^{2}\gamma }{V^{2}}\right)\exp\left(-n^{2}\gamma /V^{2}\right) \end{array}$$
(Benedict-Webb-Rubin [1]). This equation of real gas contains eight constitutive constants
$$\left(R,B_{0},A_{0},C_{0},b,a,\alpha ,\gamma \right)$$
which have to be adapted with measurement. The following questions arise: Why not seven or nine constants? Is there a theoretical background for understanding this tricky constitutive equation? Are there principles for constructing constitutive equations, i.e., the so-called material axioms?

To answer these questions, we have to start from the beginning, specifically the *balance equations* of phenomenological thermodynamics (statistical and stochastic thermodynamics are out of scope here), which include the following [2,3,4]: the balance equations of mass, momentum, energy, moment of momentum, entropy, and some more balance equations depending on the considered complex material and the thermodynamic theory applied. These balance equations are differential equations of the basic (wanted) fields, which contain differential operators, usually gradients and time derivatives operating on space and time, representing a field formulation of non-equilibrium thermodynamics. The balance equations are not determined because more fields appear than balance equations. For example, in the five balance equations of mass, momentum and internal energy, it not only contains the wanted five fields (mass density, material velocity, and internal energy) appear, but 13 additional fields emerge: Cauchy stress tensor, external forces, heat flux density, and internal energy supply. Consequently, additional material-dependent field equations are needed, i.e., the so-called *constitutive equations* that are defined based on the *state space* (constitutive space), which is spanned by the basic fields and their derivatives [5]. Thus, a *constitutive mapping* is implemented with the state space as its domain and the constitutive fields as its range. Inserting the constitutive fields into the balance equations, the *balance equations on the state space* are generated by applying the chain rule. These balances on state space represent a complete system of differential equations that are tied together with the *dissipation inequality* representing the second law of the entropy balance. All constitutive equations on the chosen state space, which jointly satisfy the balance equations and the dissipation inequality, form the *class of materials* [6].

Now the following question arises: how do we solve the balance equations by taking into consideration the dissipation inequality [7]? Do we use the cumbersome procedure of first solving the balance equations and look after the dissipation inequality or generate a class of materials that satisfies the dissipation inequality by construction? Here, the second case is considered by using an easy toy model that enlightens the current understanding.

Because the differential operators of the balance equations on state space are linear, these balance equations are linear in the so-called *higher derivatives*, which are the time and position derivatives of the state space variables, which are not included in the state space itself [6]. Consequently, the higher derivatives represent *process directions* in the state space. Using the evident but nevertheless axiomatic statement, “process directions which are in the kernel (null-space) of the balance equations do not appear in the entropy production”, the *no-reversible-direction axiom* can be established [8,9]. The local entropy production does not depend on the process directions. Introducing a so-called *Lagrange multiplier*, the balance equations can be directly connected to the dissipation inequality, resulting in the *Liu relations*, which do not include the process direction anymore [10]. The Liu relations represent the constraints for the constitutive equations with regard to the dissipation inequality, which can be exploited without taking process directions into account. How this sketched program works in detail is elucidated in this short note.

## 3. Balance Equations on State Space

Local *balance equations* can be written in different shapes
(2)$$\frac{\partial }{\partial t}\left(\rho \Psi \right)+\nabla \cdot \left(\rho v\Psi +\Phi \right)+\Sigma =0\;⟶\;\partial_{t}u_{A}+\partial_{j}\Phi_{A}^{j}=r_{A},\;A=1,2,\ldots ,M.$$
Here, Ψ or $u_{A}$ is the balanced basic quantity, which may be of an arbitrary rank; ρ the mass density, Φ is the non-convective flux or $\Phi_{A}^{j}$ the convective flux of Ψ; ***v*** is the material velocity; and Σ or $r_{A}$ is the sum of production and supply of Ψ. One of the *M* equations is that of the entropy A=S, which is expressed as follows:(3)$$u_{S}\;=\;\rho s,\;\Phi_{S}^{j}\;=\;v^{j}\rho s+\Xi^{j},\;r_{S}\;=\;\gamma +\sigma \;\geq \;\gamma$$
where the specific entropy is represented as *s*, the non-convective entropy flux as $\Xi^{j}$, the entropy supply density as γ, and the entropy production density as σ≥0. The inequality in (Equation 3) characterizes the second law.

The fields in (Equation 2)${}_{2}$ and (Equation 3) depend on the considered material, which is characterized by a *state space* Z∋z, which is spanned by its components ***z***. These material dependent fields are the following *constitutive equations*: (4)$$u_{A}\;=\;u_{A}\left(z\right),\;\Phi_{A}^{j}\;=\;\Phi_{A}^{j}\left(z\right),\;r_{A}\;=\;r_{A}\left(z\right).$$
which are inserted into the balance equations (Equation 2)${}_{2}$ and the *dissipation inequality* (Equation 3)${}_{3}$. Applying the chain rule results in the following *balance equations on state space*: (5)$$\begin{array}{ccc} \frac{\partial u_{A}\left(z\right)}{\partial z}\partial_{t}z+\frac{\partial \Phi_{A}^{j}\left(z\right)}{\partial z}\partial_{j}z & = & r_{A}\left(z\right), \end{array}$$(6)$$\begin{array}{ccc} \frac{\partial u_{S}\left(z\right)}{\partial z}\partial_{t}z+\frac{\partial \Phi_{S}^{j}\left(z\right)}{\partial z}\partial_{j}z & \geq & \gamma (z) \end{array}$$
which are linear in the *higher derivatives* (${}^{⊤}$ means transposed, and := marks a setting).
(7)$$y^{⊤}\;:=\;\left(\partial_{t}z,\partial_{j}z\right).$$
The following matrices are introduced: (8)$$\begin{array}{ccc} A\left(z\right)\;:=\;\left(\frac{\partial u_{A}\left(z\right)}{\partial z},\frac{\partial \Phi_{A}^{j}\left(z\right)}{\partial z}\right), & \; & B\left(z\right)\;:=\;\left(\frac{\partial u_{S}\left(z\right)}{\partial z},\frac{\partial \Phi_{S}^{j}\left(z\right)}{\partial z}\right), \end{array}$$(9)$$\begin{array}{ccc} C\left(z\right)\;:=\;r_{A}\left(z\right),\; & \; & D(z)\;:=\;\gamma (z), \end{array}$$
The balances on state space (Equation 5) and (Equation 6) write the following: (10)A(z)y=C(z),B(z)y≥D(z).
$u_{A}\left(z(t,x)\right)$ are the wanted basic fields, whereas $\Phi_{A}^{j}\left(z(t,x)\right)$ and $r_{A}\left(z(t,x)\right)$ are given as the functions of the basic fields or from the environment of the considered system.

## 4. Material Axioms

The constitutive equations (Equation 4) generating the balance equations on state space (Equation 10) cannot be arbitrary. (The funny materials of Disney movies do not exist in the real world.) A(z), C(z), B(z), and D(z) have to satisfy material axioms which describe the constraints that materials have to obey. These axioms include the following:1.The second law (Equation 10)${}_{2}$ and its additions;2.Transformation properties by changing the observer;3.Material symmetry;4.State spaces Z that guarantee the finite speed of wave propagation.

Does the expert know if the real gas (Equation 1) satisfy these axioms, or do they ignore them? Here, we are interested in #1. More hints can be found in the literature [3,6].

## 5. Process Directions

The state space variables z(t,x) depend on time and position according to (Equation 5) and (Equation 6). For each event (t,x), the higher derivative (Equation 7) represents the *process directions in the state space Z.* A process is here defined by the wanted (basic) fields $u_{A}\left(z\left(t,x\right)\right)$, which satisfy the balance equation (Equation 5) and the dissipation inequality (Equation 6). Having solved the balance equations (Equation 10)${}_{1}$ (or (Equation 5)) for given initial conditions and geometrical constraints, we know the higher derivatives $y^{⊤}\left(t,x\right)$, and we have to check if these higher derivatives satisfy the dissipation inequality (Equation 10)${}_{2}$ or not so that the balance equation (Equation 10)${}_{1}$ can be attached to a thermodynamical process, depending on the chosen constitutive equations A(z), C(z), B(z) and D(z). This procedure is cumbersome because if the dissipation inequality (Equation 10)${}_{2}$ is not satisfied by the chosen constitutive equations, we have to start the procedure again with renewed constitutive equations until (Equation 10)${}_{2}$ is satisfied, i.e., the so-called *global procedure*.

Does a more specific procedure exist that can determine the possible constitutive equations in advance without solving (Equation 10)${}_{1}$ so that (Equation 10)${}_{2}$ is satisfied? To answer this question, the space of the higher derivatives (of all process directions) at an arbitrary but fixed event $\left(t_{0},x_{0}\right)$ is introduced.
(11)$$Y\left(t_{0},x_{0}\right)\;∋\;y^{⊤}\left(t_{0},x_{0}\right).$$
This means the global consideration of balance equation (Equation 10)${}_{1}$ is changed into a *local procedure*, resulting in two following statements (I and II), which exclude each other as follows:I.All local solutions (irreversible and reversible ones) of the local balance equation (Equation 10)${}_{1}$ satisfy the dissipation inequality (Equation 10)${}_{2}$.
(12)$$\begin{array}{ccc} \underset{k}{⋀}\left\{y^{k}\in Y_{>}\left(t_{0},x_{0}\right)|Ay^{k}=C\right\} & ⟶ & \left\{By^{k}>D\right\}, \end{array}$$
(13)$$\begin{array}{ccc} \underset{m}{⋀}\left\{y_{eq}^{m}\in Y_{=}\left(t_{0},x_{0}\right)|Ay_{eq}^{m}=C\right\} & ⟶ & \left\{By_{eq}^{m}=D\right\}. \end{array}$$
The process directions are divided into irreversible $y^{k}$ and reversible $y_{eq}^{m}$. The linear combined process direction, 1>α>0,
(14)$$A\left(\alpha y^{k}+\left(1-\alpha \right)y_{eq}^{m}\right)\;=\;C\;⟶\;B\left(\alpha y^{k}+\left(1-\alpha \right)y_{eq}^{m}\right)\;>\;D$$
satisfies the balance equations and belongs to an irreversible process according to (Equation 14)${}_{2}$ and (Equation 12)${}_{2}$. No additional reversible process directions belonging to $Y_{=}\left(t_{0},x_{0}\right)$ can be created by a linear combination of $y^{k}$ and $y_{eq}^{m}$.

II.According to (Equation 12) and (Equation 13), there are local solutions of the balance equations (on state space) $y^{k}$ and $y_{eq}^{m}$, which satisfy the dissipation inequality. Additional process directions $y_{□}^{j}$ are now presupposed, which do not satisfy (Equation 12) and (Equation 13), representing local solutions of the balance equations, which do not satisfy the dissipation inequality (Equation 10)${}_{2}$.

(15)$$\underset{j}{⋀}\left\{y_{□}^{j}\in Y_{<}\left(t_{0},x_{0}\right)|Ay_{□}^{j}=C\right\}\;⟶\;\left\{By_{□}^{j}<D\right\}.$$
By presupposing (Equation 15), the space of the higher derivatives (Equation 11) is expressed as follows: (16)$$Y\left(t_{0},x_{0}\right)\;=\;Y_{>}\left(t_{0},x_{0}\right)\vee Y_{=}\left(t_{0},x_{0}\right)\vee Y_{<}\left(t_{0},x_{0}\right).$$
As proved in Section 6, from statement #II, it follows that additional reversible process directions beyond those in $Y_{=}\left(t_{0},x_{0}\right)$ can be created from $Y_{>}\left(t_{0},x_{0}\right)\vee Y_{<}\left(t_{0},x_{0}\right)$. This strange result paves the way to the axiom of no-reversible process directions.

## 6. Reversible Process Directions in Non-Equilibrium

From (Equation 12)${}_{2}$ and (Equation 15)${}_{2}$, the following occurs: (17)$$By^{k}>D>By_{□}^{j}\;⟶\;By^{k}-D\;>\;0\;>\;By_{□}^{j}-D.$$
The inequality (Equation 17)${}_{2}$ is transformed into an equality by introducing two positive constants.
(18)$$\alpha >0,\;\beta >0:\;By^{k}-D+\alpha \left(By_{□}^{j}-D\right)\;≐\;By_{□}^{j}-D+\beta \left(By^{k}-D\right).$$
This results in the following: (19)$$By^{k}\left(1-\beta \right)+By_{□}^{j}\left(\alpha -1\right)\;=\;D\left(-\beta +\alpha \right).$$
Setting
(20)−β+α≐1⟶α−1=β
creates a reversible process direction
(21)$$B\left(\left(1-\beta \right)y^{k}+\beta y_{□}^{j}\right)\;=\;D$$
which is, according to (Equation 12)${}_{1}$ and (Equation 15)${}_{1}$, also a local solution of the following balance equations: (22)$$A\left(\left(1-\beta \right)y^{k}+\beta y_{□}^{j}\right)\;=\;\left(1-\beta \right)C+\beta C\;=\;C,\;0<\beta <1,\;\left(2>\alpha >1\right).$$
Consequently, if not all local solutions of the balance equations satisfy the dissipation inequality (#II), a reversible process direction (Equation 21) can be constructed via the linear combination of non-reversible ones ($Y_{>}\left(t_{0},x_{0}\right)\vee Y_{<}\left(t_{0},x_{0}\right)$), which is a local solution of balance equation (Equation 22). This strange result is discussed in Section 7.

## 7. Coleman–Mizel’s Shape of the Second Law

The dissipation inequality (Equation 10)${}_{2}$ representing the second law can be differently interpreted as follows:(#I)If all process directions satisfy the dissipation inequality, arbitrary constitutive equations are not possible, and they are restricted by the dissipation inequality.(#II)If the constitutive equations A(z), C(z), B(z) and D(z) are given, the dissipation inequality (Equation 10)${}_{2}$ excludes those process directions, which do not satisfy balance equation (Equation 10)${}_{1}$.The second law states nothing about these two excluding cases. Consequently, an amendment to the second law is required for deciding which case, #I or #II, is valid. Here, this decision is given by an axiom which excludes the situation described in Section 6 as follows:(23)$$\begin{array}{c} A\mathrm{reversible}\mathrm{process}\mathrm{direction}\;\\ \mathrm{cannot}\mathrm{be}\mathrm{generated}\mathrm{by}\mathrm{non}-\mathrm{reversible}\mathrm{ones}. \end{array}$$
Therefore, the second inequality of (Equation 17)${}_{1}$ must not be valid, which means
(24)$$Y_{<}\left(t_{0},x_{0}\right)\;=\;\emptyset ,\;\mathrm{for}\;\mathrm{all}\left(t_{0},x_{0}\right)$$
is valid. Consequently, process directions of negative entropy production do not exist. Consequently, all process directions are of non-negative entropy production. In particular, there are no solutions of the balance equations of negative entropy production. Therefore, #I, (Equation 12) and (Equation 13), is true.
(25)$$\begin{array}{ccc} & & \mathrm{All}\mathrm{local}\mathrm{solutions}\mathrm{of}\mathrm{the}\mathrm{balance}\mathrm{equations}\\ & & \;\mathrm{have}\mathrm{to}\mathrm{satify}\mathrm{the}\mathrm{dissipation}\mathrm{inequality}. \end{array}$$

After having excluded via (Equation 24) all the process directions of negative entropy production, the dissipation inequality represents a constraint with regard to constitutive equations (Equation 8) and (Equation 9) and is not excluding any process direction of $\left(Y_{>}\left(t_{0},x_{0}\right)\vee Y_{=}\left(t_{0},x_{0}\right)\right)$, which satisfy the second law. Consequently, the constitutive equations cannot be independent of each other but must have the property that the entropy production density is not negative for all local, and therefore also for all global, solutions of the balance equations. This is the Coleman–Mizel (CM) formulation of the second law [11], which presupposes the validity of (Equation 12) and (Equation 13) ad hoc. Taking (Equation 24) into account, the CM formulation of the second law follows and adopts its physical interpretation; there are no solutions of the balance equations of negative entropy production.

## 8. Entropy Production

Taking a special material into consideration for which the process-direction-independent A,C,B and *D* are given and for which (Equation 25) is valid. According to (Equation 14), a lot of processes of different process directions are possible, resulting in the fact that the matrix A of the constitutive equations has a kernel K, $\left(Y_{>}\left(t_{0},x_{0}\right)\vee Y_{=}\left(t_{0},x_{0}\right)\right)⊃K∋y_{ker}$
(26)$$Ay_{ker}\;=\;0\;⟶\;y\;=\;y_{0}+y_{ker}\;⟶\;Ay\;=\;Ay_{0}\;=\;C.$$
Here, *y* and $y_{0}$ are local solutions of the balance equations according to (Equation 10)${}_{1}$ and (Equation 26)${}_{3}$, and with them the following also satisfies the balance equations: (27)$$A\left(\alpha y+\left(1-\alpha \right)y_{0}\right)\;=\;C$$

Introducing the entropy production σ≥0, the dissipation inequality becomes the following, according to (Equation 12): (28)$$By\;=\;B\left(y_{0}+y_{ker}\right)\;=\;D+\sigma \left(y\right)\;=\;D+\sigma \left(y_{0}+y_{ker}\right)\geq \;D.$$
According to (Equation 26)${}_{1}$, the kernel of A is not present in the balance equations. Therefore, it should not be present in the entropy production.
(29)σ(y)=σ(y0+yker)=axσ(y0).
Although clear from the view of physics, (Equation 29)${}_{2}$ is an axiom that is more stringent than the verbal formulations (Equation 23) and (Equation 25), as demonstrated below in (Equation 34). Consequently, (Equation 28)${}_{2}$ and (Equation 29)${}_{2}$ gives the following: (30)$$By-By_{0}\;=\;D+\sigma \left(y\right)-\left(D+\sigma \left(y_{0}\right)\right)\;=\;0\;=\;B\left(y-y_{0}\right)\;=\;By_{ker},$$
where *B* is perpendicular to the kernel of A.

Consider the two arbitrary local solutions of the balance equation (Equation 26)${}_{3}$, $y^{1}$ and $y^{2}$, and (Equation 26)${}_{1,2}$.
(31)$$0\;=\;A\left(y^{1}-y^{2}\right)\;=\;A\left(y_{0}^{1}-y_{0}^{2}\right)\;⟶\;y_{0}^{1}-y_{0}^{2}\;=:\;y_{ker}^{12}\;\in \;K,$$
The below equation is given according to (Equation 26)${}_{2}$, (Equation 31)${}_{3}$, and (Equation 30)${}_{4}$.
(32)$$\begin{array}{ccc} y^{1}-y^{2} & = & y_{ker}^{12}+y_{ker}^{1}-y_{ker}^{2}, \end{array}$$
(33)$$\begin{array}{ccc} B(y^{1}-y^{2}) & = & B\left(y_{ker}^{12}+y_{ker}^{1}-y_{ker}^{2}\right)\;=\;0\;=\;\sigma \left(y^{1}\right)-\sigma \left(y^{2}\right), \end{array}$$
The above equation is formulated according to (Equation 28)${}_{2}$. Consequently, the entropy production does not locally depend on the process direction according to (Equation 33)${}_{3}$. If one process direction at $\left(t_{0},x_{0}\right)$ is reversible, specifically $y_{rev}\in Y_{=}\left(t_{0},y_{0}\right)$, all other process directions at $\left(t_{0},x_{0}\right)$ are also reversible, and an equilibrium state is present. This results in the verbal formulation of the axiom of no-reversible process directions, which represents an amendment to the second law
(34)$$\begin{array}{ccc} & & \;\mathrm{Except}\mathrm{in}\mathrm{equilibria},\\ & & \mathrm{reversible}\mathrm{process}\mathrm{directions}\mathrm{in}\mathrm{state}\mathrm{space}\mathrm{do}\mathrm{not}\mathrm{exist}. \end{array}$$

## 9. Liu Relations

Introducing a suitable matrix λ, which does not depend on the process direction *y*, and via (Equation 28)${}_{2}$ and (Equation 26)${}_{3}$, the following inequality is valid: (35)(B−λA)y=D+σ(y)−λC≥D−λC.
According to (Equation 26)${}_{1}$ and (Equation 30)${}_{4}$, λ connects A with *B*, and the *Liu relations* are generated by setting the following, which are independent of the process directions:(36)B≐λA⟶λC≥D

Attention must be paid to the following argumentation: It is **not** said that the bracket in front of *y* in (Equation 35) has to vanish because higher derivatives are not in the state space and, therefore, independent variables, a procedure that can be found in [12,13] and is not very convincing. Here, a formal argumentation is used; there exists a matrix λ, which connects the matrices ***B*** and A according to (Equation 36)${}_{1}$. Of course, the bracket in front of ***y*** in (Equation 35) vanishes from this setting, but the argumentation is not based on the higher derivatives.

To check that the types of matrices in (Equation 36) are well matched with λ, see the following table, which shows the corresponding types of the different matrices.
matrixshape rows-columnsaccording to*D*1-1(Equation 9)_2_yL-1(Equation 7)B1-L(Equation 10)${}_{2}$CK-1(Equation 9)_1_**A**K-L(Equation 10)_1_λ1-K(Equation 36)_1_

Because A has no right-hand inverse, λ is not determined via the given ***B*** and A, but the constitutive quantities ***B*** and A are not independent of each other. There exists a λ so that the Liu relations (Equation 36) are satisfied. A tutorial example concerning the use of the Liu equations can be found in [6].

The Liu relations represent constraints for the constitutive properties. They bring together the constitutive quantities of the balance equation with those of the dissipation inequality. These restrictions have to be inserted into the balance equations and the dissipation inequality (Equation 10), resulting in a differential equation, which takes the dissipation inequality into account.
(37)By=λC.

Another kind of exploitation is that the dissipation inequality regards the λ as a Lagrange multiplier introducing the balance equations into the dissipation inequality. The dissipation inequality (Equation 35)${}_{2}$ can be replaced with the following: (38)Bω−λ(Aω−C)=D+σ≥D.
Here, ω is an arbitrary process direction, which does not necessarily have to satisfy the balance equations and/or the dissipation inequality (Equation 12) and/or (Equation 13). The usual dictum is adding the balance equations as constraints with a Lagrange factor to the dissipation inequality, resulting in arbitrary process directions. This procedure is justified by the Liu relations (Equation 36). The row λ, introduced by (Equation 36)${}_{1}$ as connection between the balance equations and the dissipation inequality, proves to be a Lagrange factor.

## 10. Conclusions

Avoiding the global procedure of solving the balance equations in consideration of the dissipation inequality globally and replacing it with the local procedure, the following question arises: how do we handle the process directions? One possibility is the above-discussed Liu procedure. One other possibility is the Coleman–Noll technique [12,13], which uses the Clausius–Duhem inequality as a dissipation inequality, replacing the entropy balance equation (Equation 3) or (Equation 6). The Coleman–Noll technique gets rid of the process directions by setting the coefficients belonging to them to zero, thus enforcing that the balance equations fall into parts that do not contain any process directions.

Reversible process directions appear only in equilibrium. The entropy production in non-equilibrium does not depend on the process direction; it is a state function. Consequently, no-reversible process directions exist in non-equilibrium. Otherwise, the entropy production becomes dependent on the process direction, and the Liu relations are not valid. These results are based on the obvious, but the axiomatic fact remains that process directions do not appear in the balance equations and do not appear in the entropy production.

## Data Availability

Not applicable.

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
