# Peer review of "Second Law and Its Amendment: The Axiom of No-Reversible Directions Revisited"

_entropy, 2023, doi:10.3390/e25081226_

Round 1

Reviewer 1 Report

The paper deals with a tutorial short-note to describe how to achieve the non-reversible directions axiom (amendment of the second law) in non-equilibrium thermodynamical. A toy-model is used to discuss the current topic of the material theory regarding the way to reached the compatibility of the constitutive equations with the dissipation inequality (representing one of the formulations of the thermodynamic second law). The question regarding the way to solve the balance equations taking into account the dissipation inequality is discussed.

In particular, the case where a class of materials generated by means of construction satisfying the dissipation inequality is investigated. Then, starting from the balance equations on state space the process directions are studied. By using a toy-model it is seen that reversible directions appear only in equilibrium, the local entropy production in non-equilibrium does not locally depend on the process directions, the axiom of no-reversible process directions, that is an amendment of the second law, can be expressed in the form: "Except in equilibria, reversible process directions in state space do not exist". Indeed, the introduction of Lagrange multipliers connecting the balance equation to the dissipation inequality lead to Liu relations, that are constraints for the constitutive equations with regard to the dissipation inequality, can be utilized without considering process directions.

  On the other hand the results agree with the axiomatic fact that process directions which are not present in the balance equations are also not present in the production of entropy. The paper is consistent with the objectives of ENTROPY.  The subject of the article is of current scientific interest. The paper deals with  a clearly written formulation of  a revisitation of second law and its amendment. 

It  is put into   in relation to previous works by the same author and by others, and the references are appropriate and necessary.  

The scope of this article  is adequately defined and justified. The  presentation  of the  problem  treated is given at a proper level of detail.

 I strongly recommend the publication of this article on ENTROPY.

Author Response

Thanks to the reviewer for his positive report. The last version of the manuscript is attached.

Reviewer 2 Report

While the general topic of the article is interesting, it requires a major revision before being publishable for several reasons:

1) The article is not very well written, it contains many grammatical errors and incomplete sentences. The language should be improved.

2) Many variables used in equations are not explained anywhere.

3) Given that this is supposed to be a tutorial, it is written in a way that is way too complicated for beginners to understand. I find it really difficult to follow. Concepts are not adequately introduced and the notation is difficult.

4) Why does the state space have to guarantee a finite speed of wave propagation? Is the standard heat conduction or diffusion equation not an adequate description of materials? (I know that relativistic effects in principle require a finite propagation speed, but this is nothing that matters for most of materials science.)

6) It should be explained in more detail why constructing a reversible process direction out of non-reversible ones is problematic. It is well known that whether or not a certain process is reversible can depend on the level of description.

7) Similarly, I do not see why reversible process directions should not be possible except in equilibria. Consider a system that is overall in nonequilibrium but in equilibrium within a certain region, reversible processes should then certainly be possible within that region. Similarly, general nonequilibrium processes are in general a superposition of reversible and nonreversible contributions (see, for instance, the GENERIC formalism).

8) The final section should be called "Conclusion", not "Closure".

In the present form, I do not think that the article is suitable for publication in Entropy.

The quality of English language should be substantially improved before the article is publishable.

Reviewer 3 Report

Dear Sirs of the Editorial Office,

please see the report attached.

Best regards.

vito Antonio Cimmelli

Author Response

entropy-248644Reply3
Reply to reviewer 3

1. The author refers the Coleman-Mizel approach to second law, introduced by those authors in 1964. (Ref. [11]). It is not clear to the referee why the author does not mention the fundamental paper by Coleman and Noll: B. D. Coleman and W. Noll, The thermodynamics of elastic materials with heat conduction and viscosity , Arch. Rational Mech. Anal. 13, 167-178 (1963), published one year early, where the same point of view is formulated. In this paper, after defining the admissible processes and deriving the local form of second law of thermodynamics, namely, the Clausius-Duhem inequality, the authors claim:
We then demand that the Clausius-Duhem inequality hold for all the admissible processes. This requirement places restrictions on the constitutive assumptions. Then, they develop a rigorous mathematical procedure for the exploitation of second law, namely, the celebrated Coleman-Noll procedure. Such a pioneering work should be included in the list of references and its contents should be commented in deep.

The reference according to Coleman-Noll is added as [12] which cannot ”be commented in deep” in a short note. The Coleman-Noll procedure is based on the Clausius-Duhem inequality for which constitutive presuppositions of entropy flux and entropy supply are necessary [13] on the contrary to Liu’s procedure which is sketched in this short note. Independed of the exploitation of the dissipation inequality is the Coleman-Mizel formulation of the 2nd law [11] and eqs. (12) and (13).

2. Page 2: The statement ”the balance equations of mass, momentum, of energy,
moment of momentum, of entropy and some more balances depending on the considered complex material” should be modified in ”the balance equations of mass, momentum, of energy, moment of momentum, of entropy and some more balances depending on the considered complex material and on the thermodynamic theory applied”.

is done

3. Page 2: The statement ”The balance equations are not determined because more fields appear than balance equations are” could be better illustrated by the simple example of continuum thermo-mechanics, wherein one has 5 balance laws and 14 unknown fields. Hence, to close the system, 9 constitutive equations for the Cauchy stress and for the heat flux are needed.

is done

4. Page 2: The author names the higher derivatives process directions. Such a name, clearly inspired by classical mechanics, would require a precise definition of thermodynamic process which, in such a case, cannot coincide with that given by Coleman and Noll and by Coleman and Mizel (see Ref [8], page 1118, properties 1). . . 8)). Such a definition should be given and the difference of the definitions should be remarked.

It is explained that the higher derivatives are equivalent to the term ”process directions on the state space”. A juxtaposition with respect to Coleman, Noll and Mizel is beyond this short note.

5. Page 3: A short description of the Liu procedure is given. The reader could take advantage of a similar description of the Coleman-Noll procedure.

I would be delighted, if the reviewer will contribute to this topic in our next joint paper.

6. Page 4: The author mentions the following constitutive axioms: 1. The second law and its additionals, 2. Transformation properties by changing the observer, 3. Material symmetry, 4. State spaces Z that guarantee finite speed of wave propagation. Indeed, the requirement 3. is not a constitutive axiom but a mathematical property of the functions describing materials with given symmetries. Moreover, the requirement 4. is postulated only in Rational Extended Thermodynamics and not in the other thermodynamic theories.
The referee suggests to eliminate the axiom 3. and clarify that the requirement 4. is not generalized.

I do not agree with the reviewer’s comments to #3 and #4: Material symmetry is a constraint for the constitutive equations, thus belonging to the material axioms which the constitutive equations have to obey. That is a severe shortcoming not to choose a state space which excludes infinite propagation speed. Consequently, #4 represents an axiom which state spaces as a domain of the constitutive have to obey.

7. Page 8: Once the theoretical framework has been clarified and the amendment to the second law has been enunciated, the way to exploit second law has been paved. Then, in Sec. 8, the author shows how the exploitation can be obtained by illustrating the Liu procedure. The referee is convinced that in Sec. 8 a similar explanation should be devoted to the Coleman-Noll procedure because: i) it was the first mathematical procedure, published 9 years before the Liu one; ii) it is more intuitive, because it does not introduce the artificial concept of Lagrange multipliers, whose physical meaning is often not easy to understand; iii) it leads to the same results of the Liu procedure, as proved in Ref. [12].

As shown in this paper, the Lagrange multiplier is not ”an artificial concept”, but can be proven by use of the no-reversible-directions axiom (NRDA). Concerning [13] ([12]old), the higher derivatives are regarded as independent variables in both procedures, a fact which physically is not clear and which is can be avoided by NRDA. Interesting is to apply the NRDA to the Coleman-Noll procedure.

8. Ref. [9], included in the list of references, is not cited in the paper. Moreover in
the title of Ref. [11], Exitence should be Existence.

is done
———————————————————————————————————
I want to thank the referee for his/her effort to improve the manuscript. Are you interested in investigating the Coleman-Noll procedure by use of the NRDA ?

=================================================

Round 2

Reviewer 2 Report

The author seems to have put more effort into questioning my scientific integrity than into adequately revising the manuscript. The article should not be published.

I am not a free proofreading service, so please don't ask me to be one.

Author Response

Sorry, that you had trouble with the manuscript.

I thank for your efforts.

W.M. 
